COMMUNICATIONS

# Persistence against benzalkonium chloride promotes rapid evolution of tolerance during periodic disinfection

Niclas Nordholt [1✉], Orestis Kanaris[1], Selina B. I. Schmidt[1] & Frank Schreiber [1✉]

Biocides used as disinfectants are important to prevent the transmission of pathogens, especially during the current antibiotic resistance crisis. This crisis is exacerbated by phenotypically tolerant persister subpopulations that can survive transient antibiotic treatment and facilitate resistance evolution. Here, we show that *E. coli* displays persistence against a widely used disinfectant, benzalkonium chloride (BAC). Periodic, persister-mediated failure of disinfection rapidly selects for BAC tolerance, which is associated with reduced cell surface charge and mutations in the *lpxM* locus, encoding an enzyme for lipid A biosynthesis. Moreover, the fitness cost incurred by BAC tolerance turns into a fitness benefit in the presence of antibiotics, suggesting a selective advantage of BAC-tolerant mutants in antibiotic environments. Our findings highlight the links between persistence to disinfectants and resistance evolution to antimicrobials.

---

[1] Division of Biodeterioration and Reference Organisms (4.1), Department of Materials and the Environment, Federal Institute for Materials Research and Testing (BAM), Berlin, Germany. ✉email: niclas.nordholt@bam.de; frank.schreiber@bam.de

The global rise of antimicrobial resistance is a major concern for public health. Disinfectants are an effective measure to prevent the transmission of bacterial pathogens in general, and antibiotic-resistant bacteria in particular. Disinfectants are chemicals that are used to inactivate mainly microorganisms on inanimate surfaces or in water. They are regulated as a main group of biocides in the EU or similarly in other parts of the world. Disinfectants are especially important in healthcare and animal husbandry settings, where infections with antibiotic-resistant bacteria are prevalent. The use, and with this the environmental dissemination, of biocides including disinfectants in terms of mass is estimated to be much higher than for antibiotics. For example, it has been estimated that the global mass of antibiotics sales was about 0.1–0.2 Tg[1] (a more recent, quantitative study estimated ~0.06 Tg for use in food animals in 2010, which is estimated to be double that used by humans[2]), while the total mass sales of biocides in the EU alone was 0.4 Tg in 2009, most of which was due to disinfectants[3]. Therefore, it can be hypothesized that antimicrobial biocides including disinfectants are a driver of microbial adaptation to antimicrobial substances on a global scale. This might be exacerbated by the wide and many times non-professional use of antimicrobial biocides[4–6]. Because of the connection between the mechanisms of antibiotic and biocide resistance[7–12], it is of great importance to understand the biology of microbial adaptation to biocides if we are to understand the current antibiotic crisis.

Benzalkonium chlorides (BAC) are quaternary ammonium compounds (QACs) that are widely used as active agents in disinfectants, antiseptics, and preservatives. They find application in industrial, healthcare, animal husbandry, and food production settings, but also consumer products (see[13] for an excellent review). Several studies showed examples of reduced susceptibilities to QACs, occurring in natural, clinical, and industrial isolates and, to a lesser extent, in laboratory evolution experiments[9,14–16]. Reduced susceptibility to QACs is underpinned by acquiring mutations in genes that increase QAC efflux by upregulation of inherent multidrug-efflux pumps[17] or by acquiring specialized QAC efflux pumps via horizontal gene transfer[18,19]. In addition, strains that have been evolved towards decreased susceptibility show reduced expression of porins related to reduced QAC uptake[20,21] and changes in membrane structure or composition[21,22]. QAC resistance mechanisms can confer cross-resistance to antibiotics. Thus, bacteria that were adapted to increasing levels of BAC can exhibit reduced susceptibility to antibiotics in terms of elevated minimal inhibitory concentrations (MIC)[8,23]. However, the high (>100 fold MIC) levels of adaptation to BAC and QACs in clinical and industrial isolates are rarely reached in laboratory evolution, using evolution protocols with stepwise increasing concentrations[15,24]. Therefore, it remains unknown what external conditions favor the emergence of the high resistance found in environmental isolates, and it can be hypothesized that laboratory evolution experiments with gradually increasing concentrations alone are not sufficient to unveil the full evolutionary potential for high-level resistance.

Disinfectants and antibiotics are typically applied periodically and, in many cases, at lethal concentrations. Periodic exposure to lethal concentrations of antibiotics has been shown previously to exert a strong selective pressure on increased survival, leading to the selection for tolerance[25,26]. Tolerance has been defined as the ability to *survive* transient exposure to an antimicrobial at otherwise lethal concentrations[27], including constitutive and inducible tolerance phenotypes. The tolerance phenotype is underpinned by genetic mechanisms, making tolerance accessible to evolution. Tolerance against antibiotics can act as a stepping stone for the evolution of resistance[28], which is defined as the ability to *grow* at high concentrations of an antimicrobial. A special case of tolerance, which has recently been in the spotlight of antibiotic research, is persistence. Antibiotic persistence describes the presence of a phenotypically tolerant subpopulation in an isogenic population of bacteria[29] and can facilitate the evolution of genetic tolerance and resistance against antibiotics[25,26,28,30]. Several intrinsic and extrinsic cues, such as entry into the stationary phase, are known to induce persistence via various pathways[31–35]. The hallmark of persistence is a multimodal time-kill curve when the population is exposed to lethal levels of antibiotics. These characteristic kill kinetics are frequently observed in response to many different antibiotics[25] and there are some examples of multimodal time-kill curves in response to QACs in the older literature[14,36–39]. However, to our knowledge, there are no reports that focus on persistence in the context of disinfectants and their consequences for the evolution of tolerance and resistance to disinfectants under periodic application regimes.

In this work, we show that *E. coli* forms persisters against BAC and that these persisters can facilitate the evolution of population-wide tolerance with consequences for antibiotic susceptibility. Bimodal killing kinetics in response to lethal BAC levels are caused by a tolerant persister subpopulation, which is partly comprised of antibiotic persisters, as determined by screening antibiotic persister mutants. Experimental evolution under periodic disinfection with BAC rapidly selects for tolerant genotypes with mutations in the late lipid A biosynthesis locus *lpxM*, which was previously not associated with tolerance. LpxM-mediated alterations in cell surface charge provide a mechanistic explanation for evolved BAC tolerance. Lastly, we show that evolved BAC tolerant mutants have a fitness-mediated, selective advantage over the ancestor in the presence of antibiotics, potentially facilitating selection of tolerant strains in settings in which disinfection is performed close to infected patients or animals (e.g., in hospitals and animal stables) or in which BAC and antibiotics are applied jointly.

## Results

**E. coli forms persisters against benzalkonium chloride.** First, we systematically determined that *E. coli* forms persisters against benzalkonium chloride (BAC), following the definition of persistence as per a recent consensus statement[29]. Three main criteria from the consensus statement are: (i) multimodal time-kill kinetics, (ii) persisters are phenotypic variants, i.e., resistant or tolerant mutants must be excluded as the reason for multimodal time-kill kinetics, and (iii) the kinetics and the fraction of persisters are "largely independent" of the concentration of the antimicrobial[29].

We investigated the killing kinetics of *E. coli* by conducting time-kill assays of stationary and exponentially growing cultures with 60 μM BAC, which is 3 times higher than the MIC (20 μM). For BAC, the MIC coincided with the minimum biocidal concentration (MBC), which is defined as the concentration which reduces the viable cell number by a factor 1000 after 24 hours. Stationary *E. coli* cultures showed bimodal killing kinetics, indicating a persister subpopulation (Fig. 1a). In contrast, disinfection kinetics in the exponential phase were unimodal (Fig. 1a). Thus, similar as for antibiotics, entry into the stationary phase triggers a physiological state that underlies persistence.

Multimodal kinetics can also result from the exhaustion of the antimicrobial[14,29,38,40,41]. We ruled out that BAC was exhausted to a sub-lethal level by spiking fresh cells for a second time-kill assay 20 min after the addition of BAC to a stationary phase culture. Again, the number of viable cells decreased in a bimodal fashion (Fig. 1b) and the initial rate of the second time-kill curve was significantly larger than the second rate of the first curve (0.38 min$^{-1}$ vs. 0.74 min$^{-1}$, $p < 0.01$, $n = 3$–6).

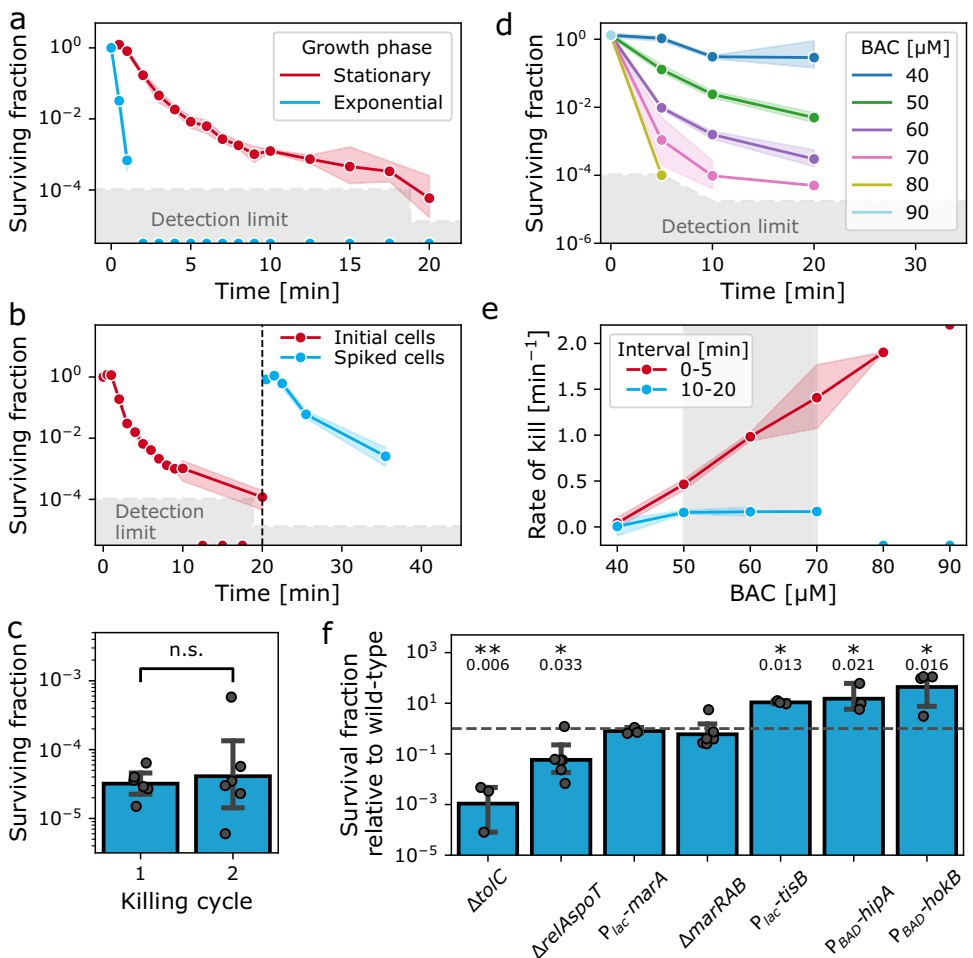

**Fig. 1 _E. coli_ forms persisters against benzalkonium chloride. a** Time kill kinetics of exponentially growing and stationary phase _E. coli_ cultures exposed to BAC at 3× MIC (i.e., 60 μM or 21 μg/ml). Points: geometric mean ± 95% C.I. ($n = 6$ biological replicates). Detection limit: 166 cfu/ml. Half circles on the $x$-axis indicate replicates with a surviving fraction below the detection limit. **b** Bimodal killing kinetics are due to a tolerant subpopulation, not due to exhaustion of biologically active BAC from the medium. Fresh cells were spiked into a time-kill assay (dashed line). Points: geometric mean, ± 95% C.I. ($n = 3$ biological replicates). **c** The persister plateau is not caused by tolerant or resistant mutants. Bars: geometric mean ± 95% C.I. ($n = 6$ biological replicates). **d** Persistence against BAC is observed in a concentration window. Points: geometric mean ± 95% C.I. ($n = 3$ biological replicates). No viable cells could be detected after 5 min of exposure above 80 μM BAC. Detection limit: 125 cfu/ml. **e** Concentration dependence of killing kinetics. The initial killing rate is correlated with the BAC concentration (Spearman correlation coefficient 0.958, $p = 2 \times 10^{-7}$. Significance of correlation: two-sided test with t-distribution of the test statistic). The second killing rate does not correlate with BAC concentration (Spearman correlation coefficient 0.31, $p = 0.416$). Points: geometric mean ± 95% C.I. as shaded area ($n = 3$ biological replicates). A lower limit estimate for the first rate at 90 μM BAC is given as a half-circle at the top of the plot. These data points were not included in correlation analysis. **f** Overlap between mechanisms that generate persisters against BAC and antibiotics. Bars: geometric mean ± 95% C.I. as error bars ($n = 3$–6 biological replicates). Δ indicates knock-out mutants; $P_{lac}$, $P_{BAD}$ indicate inducible promoter; for strain details, refer to Table S3. Significance of difference to wild-type indicated by asterisks: *$p < 0.05$; **$p < 0.01$, n.s. not significant (two-tailed unpaired t-test of log-transformed survival fraction). Exact $p$-values are indicated below the asterisks. Source data are provided as a Source Data file.

Next, we excluded that resistant or tolerant mutants are responsible for the bimodal killing kinetics. To this end, we sub-cultured the survivors of a time-kill assay and determined the persister fraction and the MIC. Neither the fraction of persisters (Fig. 1c) nor the MIC (Fig. S1) was altered.

Lastly, we determined whether the rate of killing and the fraction of persisters is only weakly dependent on the concentration of the antimicrobial. We conducted time-kill assays with different concentrations of BAC above the MBC (Fig. 1d) and found that the kill kinetics partly depend on the BAC concentration. At the lowest concentration (40 μM BAC, 2x MIC) the kinetics were unimodal as determined by comparing unimodal and bimodal fits to the data (see "Methods"). In the concentration range between 50 and 70 μM, killing kinetics were

bimodal. Here, the second killing rate was concentration-independent (Fig. 1e). At all concentrations, the initial killing rate, as well as the surviving fraction (Fig. S1), was correlated with the BAC concentration. At the two highest concentrations (80 and 90 μM BAC) we could not measure bimodal killing kinetics; however, we cannot exclude them as the fraction of surviving cells after 10 and 20 min was below the detection limit of our assay. The concentration dependence of the killing kinetics likely originates from the higher toxicity of BAC as compared with antibiotics.

In summary, we think that the phenomenon which we observe here is best described within the existing framework of persistence with the deviation that persistence against BAC is restricted to a concentration window. We expect similar results

for other biocides because they act upon multiple cellular targets which drastically increases their toxicity above certain threshold concentrations. The presence of persisters can result in unexpected failure of disinfection and, as we will show in this paper, have important implications for the evolution of bacterial defense mechanisms.

**The stringent response and drug efflux are involved in persistence against benzalkonium chloride.** Next, we investigated the mechanisms of BAC persistence and determined that persistence against BAC shares a mechanistic basis with antibiotic persistence and resistance. To this end, we screened mutants of genes that have been associated with changes in the fraction of antibiotic persisters for their ability to survive BAC treatment. Mutants in the stationary phase were exposed to BAC and the fraction of persisters after 20 min was determined by plating.

Overexpression of persistence-inducing toxins (*tisB, hokB, hipA*) increased the survival against BAC by more than 10-fold (Fig. 1f). The function of HipA and HokB requires (p)ppGpp, a global alarmone that controls the starvation-induced stringent response[42–45]. Consistent with this, a mutant that lacks both (p)ppGpp-synthesizing enzymes, *relA* and *spoT*, showed 20-fold decreased survival (Fig. 1f). These results are in line with the elevated fraction of persisters in the stationary phase (Fig. 1a), as nutrient starvation induces the accumulation of (p)ppGpp[42].

Another general defense mechanism against antimicrobials is multidrug efflux. The AcrAB-TolC-system is a major determinant of multidrug efflux in *E. coli* with a wide range of substrates, including BAC[46]. A knockout of *tolC* showed an almost 1000-fold decrease of the survival fraction (Fig. 1f), suggesting that TolC-mediated efflux is involved in BAC tolerance of the persister sub-population. This result is in agreement with previous studies that showed that heterogeneity in the partitioning of the AcrAB-TolC[47] efflux pump and heterogeneity in *tolC* expression[48] is related to antibiotic persistence. Together with observations that implicate increased efflux with resistance to membrane-disrupting QACs in *E. coli* and other species[8,17–19,49,50], our result indicates that persister cells are more tolerant to BAC due to increased efflux. However, further experiments are needed to underscore this hypothesis. Along those lines, we hypothesized that stochastic expression of the *marRAB* system, a regulator of *acrAB* and *tolC*[51], could underlie BAC persistence by causing heterogenous gene expression. However, neither overexpression of *marA* nor deletion of the *marRAB* operon did affect survival in the presence of BAC (Fig. 1f), suggesting that heterogeneous expression of *marA* does not underlie persistence against BAC.

Taken together, our data show that there is an overlap in the mechanisms that generate persisters against antibiotics and BAC. This suggests that the BAC persister subpopulation is partly composed of antibiotic persisters. These findings imply that failure of disinfection could select for antibiotic persisters. At the same time, our findings indicate that strategies against antibiotic persisters could be effective against BAC persisters[52].

**Periodic selection for BAC persisters rapidly evolves tolerant mutants.** Next, we established that persistence promotes the evolution of tolerance under periodic disinfection by conducting experimental evolution. The experimental setup included a short treatment at lethal concentrations followed by relatively long recovery periods (Fig. 2a). Similar experiments with antibiotics resulted in the evolution of genotypes with increased persister levels[25,26]. During treatment, the selection pressure is on tolerance, whereas it is on growth-rate during recovery. This profoundly differs from classical evolution experiments, where selection pressure is always on growth in the presence of antimicrobials. We monitored the dynamics of tolerance evolution by plating dilutions after 15 min of BAC exposure, when the cells were propagated to the next round, and additionally after 24 h of exposure (Fig. 2a, b). The fraction of tolerant cells after 15 min significantly increased within only three growth and killing cycles. After the 12th treatment cycle, the populations reached an average 2200-fold increase of the survival fraction, compared to the ancestor. Interestingly, the survival fraction after 24 h of BAC exposure also increased significantly, although the cells were not selected to survive for this extended period.

**Trade-off between tolerance against BAC and growth rate in evolved clones.** We phenotypically characterized isolated clones from each of the six evolved lines (S1−S6; Fig. 2b). The survival fraction after 20 min of BAC treatment increased by a factor 500–2300, while the MIC did not increase (Figs. 2c and S2). This decoupling of tolerance and resistance has been observed previously for tolerance evolution against antibiotics[26]. Importantly, these tolerant mutants would not be identified in a classical antimicrobial susceptibility screening.

Despite the selective pressure on growth rate during recovery, all clones had a 10–25% lower growth rate than the ancestor, signifying the cost of tolerance (Fig. 2c). Moreover, growth rate and tolerance were negatively correlated, suggesting a trade-off between these two traits (Fig. 2d). We developed a mathematical model to simulate the population dynamics during the evolution experiment (see "Methods" for details). Briefly, the ODE system that describes the change of the number of bacterial cells, *N*, and the resource, *R*, consists of the following equations:

$$\frac{dN}{dt} = \sum_i^n G_i \mu_i \qquad (1)$$

$$\frac{dR}{dt} = -\sum_i^n G_i \mu_i e_i \qquad (2)$$

with $\mu_i = \mu\max_i \frac{R}{R + K_m}$, where $\mu\max_i$ is the maximal specific growth rate of genotype $i$, $G_i$ is the number of cells of genotype $i$, $e_i$ the genotype-specific inverse yield (glucose/cells), and $K_m$ the resource concentration at which $\mu_i$ is at 50%. To simulate the population dynamics in the evolution experiment, the ancestor started with $10^4$ cfu/ml and the mutant with $10^{-1}$ cfu/ml (i.e., $10^5$ ancestor cells and 1 mutant in a total culture volume of 10 ml). Each growth cycle went on for 24 h after which a genotype-specific killing factor and a common dilution factor (1:100) were applied before another round of growth was started by setting the value for the common resource to the initial value. We parameterized the model with the experimentally determined parameters growth rate and tolerance and it was able to quantitatively capture the population dynamics during evolution (Fig. 2e). In all cases the mutant was able to invade from a single mutant cell and rapidly fix in the population, driving the ancestor to extinction within 5–6 days, despite a reduced growth rate of up to 25% (Fig. S3).

Taken together, the large increase in tolerance allowed the mutants to fix in the population despite a considerably decreased growth rate. The rapid fixation of the tolerant mutants illustrates the large fitness advantage over the ancestor. It also suggests that only a few incidents of incomplete disinfection due to persisters could be a threat to the efficacy of disinfection protocols.

**BAC tolerance is associated with mutations in late lipid A biosynthesis and reduced cell surface charge.** In parallel to the phenotypic characterization, we sought to identify the genetic and mechanistic basis of evolved BAC tolerance by whole-genome

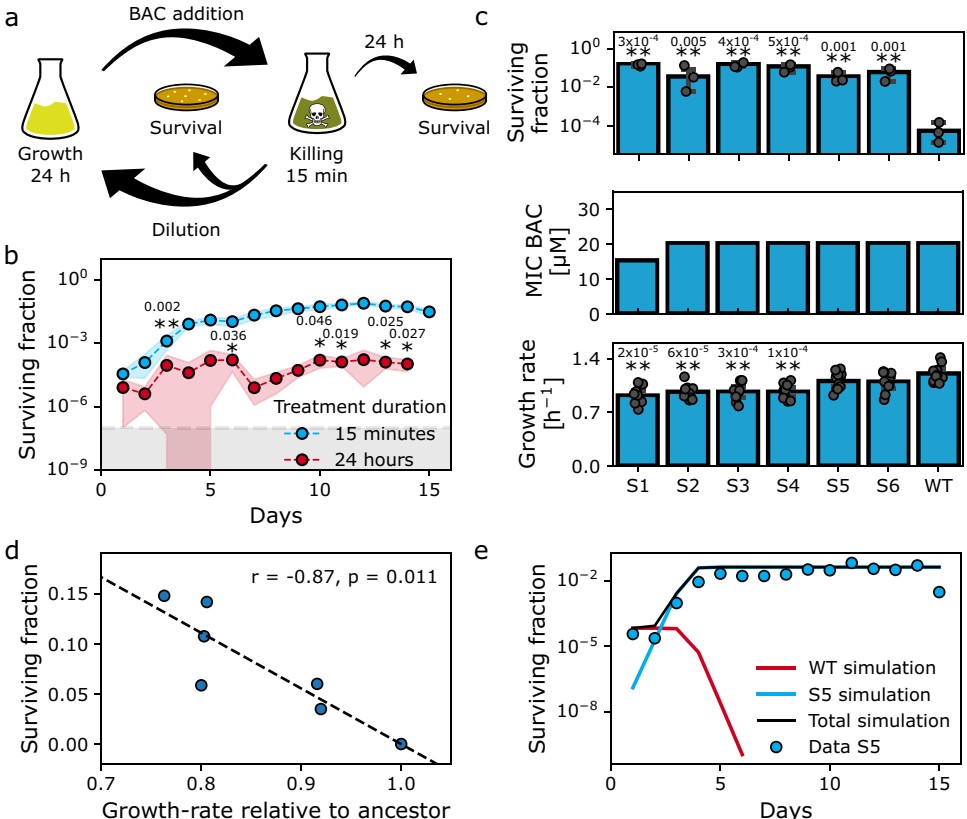

**Fig. 2 Periodic failure of disinfection rapidly selects for tolerance against BAC. a** Schematic of the evolution experiment selecting for the survival of BAC disinfection. **b** Daily exposure to BAC rapidly selects for tolerance. Shown are the geometric means ± 95% C.I., $n = 6$ biological replicates. The first round that is significantly different from the beginning of the treatment is marked by asterisks. *$p < 0.05$; **$p < 0.01$ (two-tailed paired t-test of log-transformed survival fractions). **c** Phenotypic characterization of six evolved clones. The surviving fraction of the evolved clones after 20 min of BAC treatment is increased ($n = 3$ biological replicates). Significance of difference to wild-type indicated by asterisks: *$p < 0.05$; **$p < 0.01$ (two-tailed t-test of log-transformed survival fraction). The MIC against BAC is not affected in the evolved clones ($n = 4$ biological replicates). The growth rate of the evolved clones is reduced, signifying the cost of tolerance ($n = 10$ biological replicates). Significance of difference to wild-type indicated by asterisks: *$p < 0.05$; **$p < 0.01$ (two-tailed unpaired t-test of growth rates). Error bars show 95% confidence intervals. For the surviving fraction, the geometric mean is reported. For the growth rate, the arithmetic mean is reported. For the MIC, the maximal value from four biological replicates is reported. **d** Trade-off between tolerance and growth rate in individual clones. Significance of correlation: two-sided test with t-distribution of the test statistic. The data are taken from panel (**c**). **e** Population dynamics can be explained by trade-off between growth-rate and tolerance of individual clones. Simulations of survival fraction during the serial passage experiment with alternating growth and kill cycles (lines) quantitatively reproduce the experimental data (circles). The parameters for growth rate and survival fraction are the same as for clone S5 in panel (**c**). Simulations of all individual clones can be found in Fig. S3. Source data are provided as a Source Data file. Exact p-values are indicated above the asterisks.

sequencing. All lines had mutations in *lpxM* or *lpxL* (Table 1 and Fig. 3a). Neither of these loci was previously associated to tolerance against disinfectants or antibiotics. This could be because previous studies selected on growth rather than survival, while *lpxL* and *lpxM* are directly associated to the survival of disinfection. Interestingly, no two lines had the same *lpxM* mutation (Table 1 and Fig. 3a). All mutations in the coding sequence of *lpxM* resulted in amino acid substitutions in the cytoplasmic part of the protein, downstream of the catalytically active HxxxxD-motif[53] (Fig. 3b).

Next, we wanted to understand how LpxM affects survival against BAC. LpxM catalyzes the last acylation step of lipid A, which constitutes the outer lipid layer of the outer membrane in Gram-negative bacteria and anchors the lipo-polysaccharide (LPS) matrix. Overexpression of *lpxM* did not influence tolerance against BAC (Fig. 4a). In contrast, disruption of *lpxM* resulted in a ~35-fold increase in survival. However, the effect on tolerance in the evolved strains was almost 100-fold higher than that, suggesting that a knock-out of *lpxM* is not sufficient in explaining the high levels of evolved tolerance and that additional mutations

are required for this. It is likely that a change of activity or substrate specificity of LpxM is responsible for the increased tolerance levels in the evolved mutants. Alternatively, we tested whether the effect of *lpxM* on tolerance is indirect, via induction of the σE-dependent envelope stress response. Deletion of *lpxM* or *lpxL*, membrane stress and other changes to the LPS structure are known to induce expression of σE-dependent genes[54,55]. While our data show that overexpression of σE increases tolerance against BAC, the increase is smaller than in the evolved clones (Fig. 4a).

The outer membrane surface of Gram-negative bacteria has a net-negative charge[56–58]. We hypothesized that the mutations in *lpxM* affect the charge of the outer membrane, leading to decreased absorption of the positively charged BAC. A change of membrane surface charge with unknown genetic basis was previously described in a *P. fluorescens* isolate highly resistant to BAC[16]. We determined changes in the surface charge relative to the ancestor by quantifying the adsorption of the positively charged protein cytochrome c to the cells (Fig. 4b). All but two clones (S5 and S6) adsorbed significantly less cytochrome c than

**Table 1 Mutations identified in evolved clones.**

| Strain | Genome position | Mutation | Amino acid substitution | Gene(s) | Annotation |
|---|---|---|---|---|---|
| S1 | 1,094,727 | Δ26,518 bp | n.a. | ymdE, ycdU, serX, ghrA, ycdX, ycdY, ycdZ, csgG, csgF, csgE, csgD, csgB, csgA, csgC, ymdA, ymdB, clsC, opgC, opgG, opgH, yceK, msyB, mdtG, lpxL, yceA, yceI, yceJ, yceO, solA, bssS, dinI, [pyrC] | IS3-mediated separate table |
| S1 | 1,939,467 | G→A | intergenic (−50/+70) | lpxM ← / ← mepM | Myristoyl-acyl carrier protein-dependent acyltransferase/peptidoglycan DD-endopeptidase MepM |
| S2 | 1,115,520 | G→T | A96E | lpxL ← | Lauroyl acyltransferase |
| S2 | 1,971,413 | Δ6,314 bp | n.a. | [tar], cheW, cheA, motB, motA, flhC, flhD | IS1-mediated separate table |
| S3 | 1,289,634 | T→C | C57R | rssB → | Regulator of RpoS |
| S3 | 1,938,693 | G→T | A242E | lpxM ← | Myristoyl-acyl carrier protein-dependent acyltransferase |
| S4 | 1,290,339 | C→T | Q292* | rssB → | Regulator of RpoS |
| S4 | 1,938,975 | A→C | M148R | lpxM ← | Myristoyl-acyl carrier protein-dependent acyltransferase |
| S5 | 1,108,517 | +9 bp | intergenic (−352/−34) | opgC ← / → opgG | Protein required for succinyl modification of osmoregulated periplasmic glucans/osmoregulated periplasmic glucans (OPGs) biosynthesis protein G |
| S5 | 1,938,799 | A→T | L207I | lpxM ← | Myristoyl-acyl carrier protein-dependent acyltransferase |
| S6 | 1,111,908 | Δ2 bp | coding (1822-1823/2544 nt) | opgH → | Osmoregulated periplasmic glucans (OPGs) biosynthesis protein H |
| S6 | 1,938,526 | A→C | W298G | lpxM ← | Myristoyl-acyl carrier protein-dependent acyltransferase |

Italic font indicates gene names; arrows indicate orientation on the genome; n.a. not applicable; the mutations for all sequenced clones per line can be found in Table S1. No mutations were identified in any of the 4 clones from the 2 control lines.

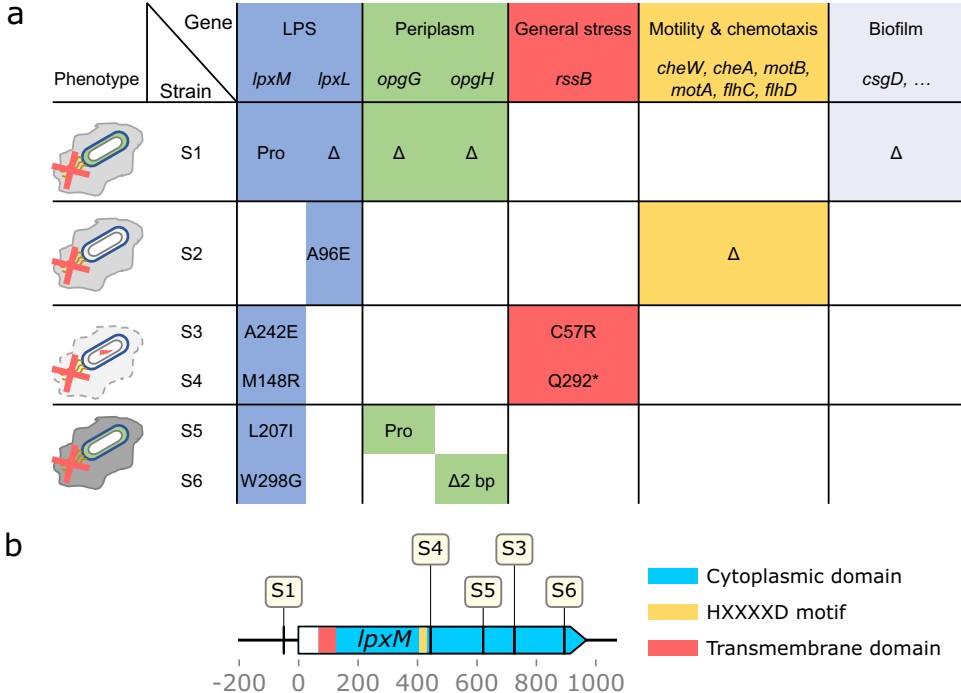

**Fig. 3 Mutations in lipid A biosynthesis and the general stress response in evolved tolerant strains. a** Mutations in genes found in the evolved clones, color-coded by cellular function in which the mutated genes are involved and their corresponding phenotype. Amino acid substitutions, deletions, and mutation in the promoter region are indicated in the table. Phenotypes are based on phenotypic characterization in Fig. S4 and color-coded according to the mutations in the table. All strains have acquired mutations in lipid A biosynthesis (blue). Strains S5 and S6 are strong biofilm formers (gray), whereas S3 and S4 show reduced biofilm formation and have acquired mutations in the regulation of the general stress response (red). All strains are less motile, although only S2 has mutations in motility genes (yellow). **b** The last gene of lipid A biosynthesis, *lpxM*, is mutated in 5 of 6 evolved lines. Mutations are labeled by clone in which they occurred. One mutation in the promoter, four mutations in the cytoplasmic domain (light blue), downstream of the functional HXXXXD acylation domain (yellow). Annotations were extracted from UniProtKB, entry P24205 (uniprot.org). No mutations were detected in the transmembrane domain (red). Only mutations from the clones analyzed in the main text are mapped (cf. Table 1 and see Table S1 for mutations in all clones).

the ancestor (Fig. 4b). Importantly, deletion of *lpxM* also results in decreased adsorption of cytochrome c. In the evolved clones, the change in surface charge showed a strong correlation with the survival fraction, which we did not observe for the *lpxM* deletion (Fig. 4c). These results indicate that, among other unknown factors, changes of the outer membrane charge due to mutations in *lpxM* contribute to high-level tolerance against BAC.

Besides the mutations in *lpxM* and *lpxL*, all clones had at least one additional loss-of-function mutation in genes involved in proteolysis of the general stress response regulator σ^S (*rssB*), synthesis of osmoregulated periplasmic glucans (*opgGH*) or deletions of genes involved in motility and chemotaxis (Table 1 and Fig. 3a). Therefore, we investigated whether biofilm formation and motility are affected in the evolved clones. We found that the effect on biofilm formation depends on the evolved clone, whereas motility is reduced in all clones (Fig. S4). This shows that biofilm-related phenotypes are not immediately related to the mutations in *lpxM*. In addition, our observation of decreased motility agrees with recent studies showing that exposure to a range of biocides negatively affects motility[59–61]. However, tolerance against BAC in a *fliC* knock-out mutant was not affected, showing that the absence of flagella or reduced motility alone does not explain the increased tolerance in the evolved clones (Fig. 4a). Moreover, even strains without mutations in motility genes were significantly less motile (Fig. S4 and Table 1). Thus, it is unlikely that reduced motility itself is contributing to BAC tolerance. Rather, our data suggests that reduced motility is a pleiotropic effect of regulatory re-wiring in the tolerant clones.

Taken together, all evolved clones have mutations in the late lipid A biosynthesis, which suggests a hitherto unknown key role of *lpxM* function in BAC tolerance. The mutations in *lpxM* contribute to a decreased net negative charge of the outer membrane and reduced adsorption of BAC. Higher levels of tolerance are then achieved by additional mutations, likely through induction of the general and envelope stress response. The high diversity of the mutations that we observed in the evolved clones suggests that the mutational target for tolerance (the 'tolerome') to BAC is large.

**BAC tolerance confers a fitness advantage in the presence of antibiotics.** Next, we demonstrated that BAC tolerance has consequences for growth, survival, and selection in the presence of bactericidal antibiotics from four different classes (β-lactams, fluoroquinolones, aminoglycosides, and antimicrobial peptides). For each of the evolved clones, we determined the MIC, the tolerance against antibiotics, and the fitness with sub-inhibitory levels of antibiotics. All clones had a two-fold increased MIC of either ampicillin, ciprofloxacin, or both (Fig. 5a), while the MIC to BAC remained unchanged (Fig. 2c). The growth-rate costs incurred by BAC tolerance (Fig. 2c) diminished in the presence of antibiotics (Fig. 5b) as determined by growth assays with antibiotic concentrations that inhibited the growth rate of the wild-type by 30–50%. Strikingly, the growth rate of several clones in the presence of ciprofloxacin was higher than that of the wildtype, indicating a selective advantage of these clones in the presence of antibiotics. We used our model to simulate competition

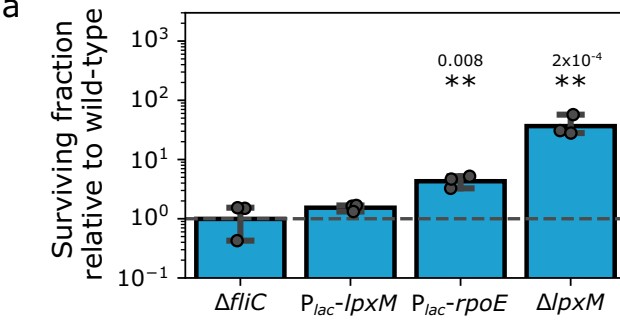

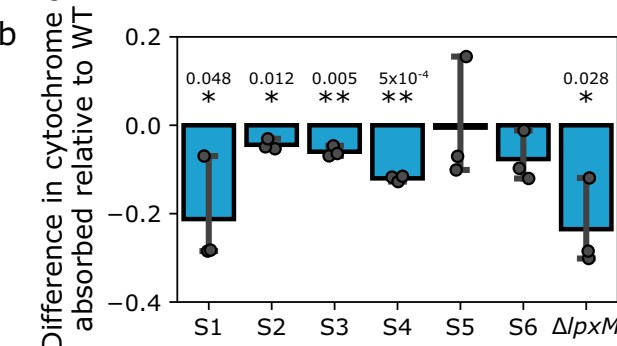

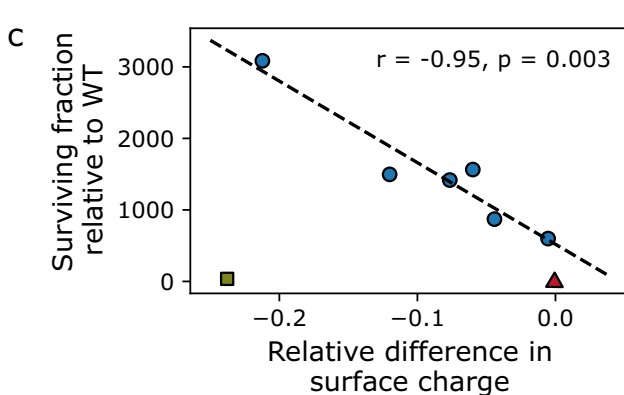

**Fig. 4 Evolved tolerance against benzalkonium chloride is related to membrane surface charge, motility, and biofilm formation. a** Mutants of outer membrane biogenesis and envelope stress, but not flagella, show increased levels of survival against BAC. Survival fractions were scaled to the survival fraction of the corresponding wild-type. Bars represent the geometric mean ± 95% C.I., $n = 4$ biological replicates. Δ indicates knock-out mutants; $P_{lac}$, $P_{BAD}$ indicate inducible promoter; for strain details, refer to Table S3. Significance of difference to wild-type is indicated by asterisks: *$p < 0.05$; **$p < 0.01$ (two-tailed t-test of log-transformed survival fraction. **b** The evolved clones and a knock-out of $lpxM$ absorb significantly less positively charged cytochrome c than the ancestor, indicating a reduction in net-negative surface charge caused by mutations in $lpxM$. Shown is the mean ± 95% C.I., $n = 3$ biological replicates. Significance of decreased cytochrome c absorption against ancestor is indicated by asterisks: *$p < 0.05$; **$p < 0.01$ (one-tailed one-sample t-test for difference from 0). **c** Quantitative relationship between tolerance against BAC and changes in surface charge in the evolved clones (circles). The wild-type is shown as triangle, the $lpxM$ knock-out is shown as square. The Pearson correlation coefficient $r$ for the evolved clones is indicated in the plot and was calculated with the data from the evolved clones only. Significance of correlation: two-sided test with t-distribution of the test statistic. Source data are provided as a Source Data file. Exact $p$-values are indicated above the asterisks.

experiments in the absence and presence of sub-inhibitory levels of antibiotics. The evolved clones were able to outcompete the wildtype in as little as seven growth cycles (Fig. S5). This novel finding could have important implications for the selection of BAC tolerant mutants in the presence of antibiotics, potentially facilitating the evolution and fixation of multi-drug resistant bacteria.

Furthermore, several clones showed cross-tolerance to antibiotics (Fig. 5b). There was a trend for increased tolerance to ampicillin (5 to 50-fold increased survival) and decreased tolerance to ciprofloxacin (7-fold decreased survival). The data show that resistance and survival are not necessarily correlated, as highlighted by two examples. First, strains S1−S4 showed a 7-fold decrease of survival at high concentrations of ciprofloxacin (Fig. 5b), despite the fitness advantage in the presence of sub-inhibitory concentrations and the elevated MIC (Fig. 5a). Second, Strain S1 did not have an altered MIC of ampicillin or gentamicin, yet survival was increased 50-fold and 5-fold, respectively. Furthermore, S1 was more susceptible to treatment with high levels of colistin, despite no change of the MIC. Thus, despite the cationic nature of both BAC and colistin, the charge of the cell surface does not affect the susceptibility of *E. coli* to these substances in the same way. These results underscore that tolerance to BAC can be acquired via different pathways, with varying consequences for antibiotic susceptibility. For example, strains S3 and S4, or S5 and S6, which acquired similar mutations, have similar phenotypes (Fig. 3a) and show similar susceptibility profiles (Fig. 5). Taken together, our data show that evolution of BAC tolerance can lead to cross-tolerance to certain antibiotics and to pronounced fitness benefits in the presence of sub-inhibitory concentrations of antibiotics. These effects on cross-tolerance have not been described before and are distinct to the readily observed cross-resistance and have important implications for scenarios in which biocides and antibiotics are used in conjunction or succession.

## Discussion

Our work provides evidence that *E. coli* forms persisters against BAC, a widely used disinfectant. This opens up three new vistas (practical, mechanistic, and evolutionary) for understanding the responses of bacteria to disinfectants. *First*, the insights provided by our data settle a historical debate in which the shape of time-kill curves in BAC disinfection has stimulated speculations about phenotypic heterogeneity in tolerance[14,38,39]. We expect that the occurrence of persisters to other disinfectants will depend on the active substance and its concentration. The relationship between disinfection kinetics, concentration, bacterial species, and persistence should be investigated in the future to establish improved disinfection protocols that safeguard the efficacy of disinfectants and diminish the risk of biocides as drivers of cross-resistance and cross-tolerance evolution. *Second*, describing the phenomenon of BAC persistence enabled us to link the mechanism of BAC survival to mechanisms that are known to generate antibiotic persisters. This approach has strong potential to facilitate understanding of mode-of-action and mode-of-tolerance studies on disinfectants in general. We expect that this understanding would greatly benefit the field that generally lacks knowledge on detailed mechanisms as compared to the antibiotic field. *Third*, knowing about BAC persistence allowed us to leverage recently established concepts of the evolution of resistance and tolerance to antibiotics from persistence[25,28]. By adapting evolution experiments for antibiotic persistence to disinfection, we found that a BAC persister subpopulation facilitates the rapid evolution of BAC tolerance under periodic dosing schemes that mimic

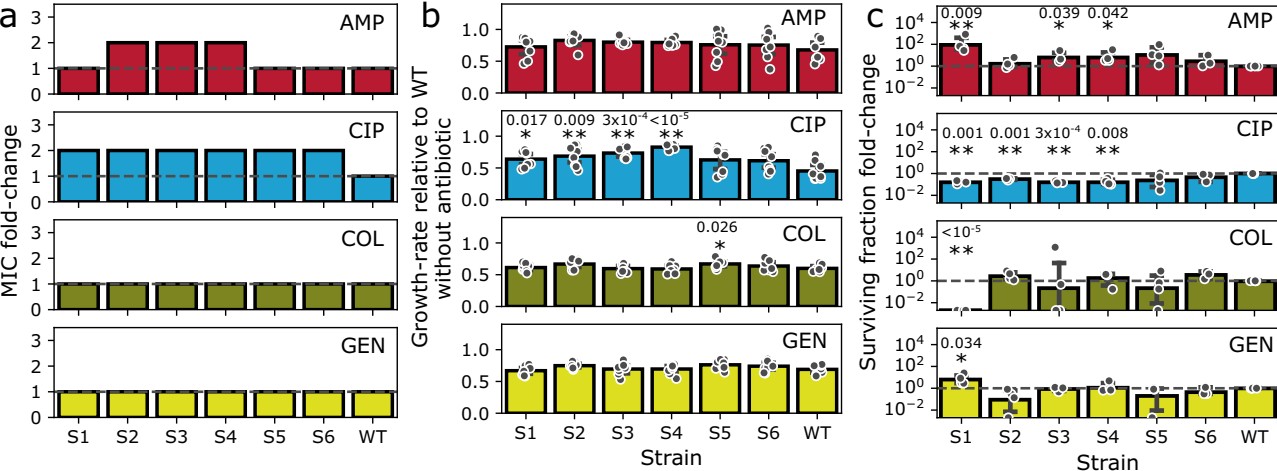

**Fig. 5 Evolution of BAC-tolerance affects susceptibility to antibiotics.** Resistance (**a**), fitness with sub-inhibitory levels of antibiotics (**b**), and tolerance (**c**) against antibiotics from the classes of β-lactams (ampicillin; AMP), fluoroquinolones (ciprofloxacin; CIP), antimicrobial peptides (colistin; COL), and aminoglycosides (gentamicin; GEN) were assessed. **a** Evolved clones show increased MICs against AMP and CIP. Bars represent the maximal value of three biological replicates. **b** Growth-rate of the clones at sub-inhibitory levels of antibiotics. Bars represent the mean ± 95% C.I., $n = 8$ biological replicates. Significant differences to the wild-type are indicated by asterisks: *$p < 0.05$; **$p < 0.01$ (two-tailed unpaired t-test of growth rate). **c** Evolution of tolerance against BAC can increase survival, but also lethality upon antibiotic stress. Bars represent geometric mean ± 95% C.I., $n = 4$ biological replicates. Significant differences to the wild-type are indicated by asterisks: *$p < 0.05$; **$p < 0.01$ (two-tailed one-sample t-test of log-transformed survival fraction fold-change for difference to 0). Half circles on the x-axis indicate replicates with a surviving fraction below the detection limit. These data points were excluded from statistical testing, except for S1 COL. Source data are provided as a Source Data file. Exact p-values are indicated above the asterisks.

disinfectant application in real world scenarios. In-depth investigations of the evolved clones identified two novel tolerance loci in the late lipid A biosynthesis (namely *lpxM* and *lpxL*) associated with a decrease of the net negative surface charge as a novel mechanism for evolved BAC tolerance providing a mechanistic explanation to observations made in natural isolates[16].

The rapid evolution of BAC tolerance in our experiment highlights the need for complementary approaches when assessing the potential for evolution of bacterial defense mechanisms to biocides. These approaches should include (i) traditional experiments selecting for growth during adaptation to increasing concentrations of biocides (including serial dilution at sub-inhibitory concentrations or selection on plates with inhibitory concentrations) and subsequent assessment of the MIC, and (ii) experiments selecting on increased transient survival, as performed in this study. Both approaches will be important to fully capture the risk associated with periodic failure of disinfection. It will be interesting to study whether the evolution of tolerance facilitates the evolution of high-level resistance to biocides, as was shown previously for antibiotics[28]. The diverse mutations and phenotypes in our evolved clones suggest a large mutational target size of tolerance (the tolerome). Previous work on antibiotics has shown that the tolerome is much larger than the resistome[62]. This discrepancy has so far not been directly addressed for disinfectants. Our study shows how the tolerome to disinfectants can be charted by selecting for survival and paves the way for future studies that investigate this underexplored genetic repertoire of bacteria.

Lastly, our data show that repeated failure of disinfection not only rapidly selects for disinfectant-tolerant mutants, but also has important consequences for tolerance and fitness in the presence of antibiotics. BAC tolerant clones can be more tolerant to lethal doses of antibiotics and be fitter in the presence of sub-inhibitory levels of antibiotics. This novel finding exemplifies that we are only beginning to understand the biology behind the adaptation

to biocides and its consequences for multi-drug resistance. The alleviation of fitness costs in the presence of antibiotics has implications for the selection and evolution of BAC tolerant clones under conditions in which cells are either transmitted from disinfected surfaces to patients and animals, or in which BAC is applied jointly with antibiotics to humans or animals (e.g., nose drops containing BAC as preservative and antibiotics to systemically fight an infection). Future studies should address whether this is the case and what are the associated consequences for the host.

Our study highlights the need for a better mechanistic understanding of the mode-of-action and the biological consequences of biocides, a deeper understanding of the biocide tolerome, and potentially stricter regulation of biocides, if we are to understand and avert the ongoing antimicrobial resistance crisis in which we heavily rely on disinfectants.

## Methods

**Bacteria.** The ancestral strain *Escherichia coli* K12 MG1655[63] was obtained from the lab of R. Mutzel. There were four mutations in the ancestor compared to the reference genome: Δ776 bp insB9–[crl], +8 bp bamD → / → raiA, +GC gltP →/← yjcO, and a ~1800 bp inversion of the P-element of prophage e14[64]. Culturing the ancestor in M9 glucose by serial passage for 150 generations was done to allow adaptive mutations to the culture medium but did not result in additional mutations. In the main text, we refer to the strain pre-cultured in M9 as the wild type or the ancestor.

Plasmids for overexpression of persister genes were transformed into the wild-type strain by electroporation. See table S3 for a complete list of strains used in this study. For overexpression analysis of *lpxM*, we used a plasmid from the ASKA library[65]. The ASKA plasmids contain the ORF of each gene under the control of an IPTG inducible promoter. A knock-out mutant of *lpxM* was obtained from the KEIO collection[66]. Mutants were compared to their respective wild-type strain from the same background. All bacterial strains used in this study are listed in Table S3.

**Culture conditions.** Unless stated otherwise, bacteria were cultured in 10 ml M9 minimal medium in 100 ml Erlenmeyer flasks with agitation at 220 rpm at 37 °C. M9 medium was composed as follows: 42 mM Na₂HPO4, 22 mM KH₂PO₄, 8.5 mM NaCl, 11.3 mM (NH₄)₂SO₄, 1 mM MgSO₄, 0.1 mM CaCl₂, 0.2 mM Uracil,

1 µg/ml thiamine, trace elements (25 µM FeCl$_3$, 4.95 µM ZnCl$_2$, 2.1 µM CoCl$_2$, 2 µM Na$_2$MoO$_4$, 1.7 µM CaCl$_2$, 2.5 µM CuCl$_2$, 2 µM H$_3$BO$_3$) and 20 mM glucose as the sole and limiting carbon source. For the persister-strain assay, one or more of the following supplements were added when needed: 1 mM IPTG to induce expression from $P_{lac}$ promoters, 24 mM arabinose to induce expression from P$_{BAD}$ promoters, 0.1 % casamino acids for strains growing poorly in M9 with glucose only (ASKA-strains, TB283, TB205, Δ$relAspoT$). When arabinose was added, glucose concentration was reduced to 10 mM. Optical density at 600 nm (OD$_{600}$) to cfu/ml conversion factors were determined by dilution and plating to be ~10$^9$ cfu/ml/OD in stationary phase and ~10$^8$ cfu/ml/OD in exponential phase.

To enumerate colony forming units (cfu), serial dilutions in phosphate buffered saline (PBS, pH 7; 10 mM Na$_2$HPO$_4$, 1.76 mM KH$_2$PO$_4$, 2.68 mM KCl, 137 mM NaCl) were plated on LB plates (Lennox formulation; 10 g/l tryptone, 5 g/l yeast extract, 5 g/l NaCl) containing 1.5 % agar. Plates were incubated at 30 °C for 16–24 h.

**Biocides and antibiotics.** Benzalkonium chloride (Sigma-Aldrich, 234427) was dissolved in sterile water to a final stock concentration of 2 mM or 0.71 g/l. We assumed an average molecular weight of BAC of 354.06 g/mol. The final concentration of BAC in the time-kill assays was 60 and 120 µM in the evolution experiment. Antibiotics that were used in this study: ampicillin (Roth, K029.4), ciprofloxacin (Sigma, 17850-25G-F), gentamicin (Sigma, G1272-10ml), colistin sulfate (Serva, 17420.02). The concentrations for the growth assays in Fig. 5b were: ampicillin 1 µg/ml, ciprofloxacin 0.005 µg/ml, colistin 0.25 µg/ml und gentamicin 0.125 µg/ml. The concentrations for the time-kill assays in Fig. 5c were: ampicillin 100 µg/ml, ciprofloxacin 1 µg/ml, colistin 10 µg/ml und gentamicin 7.5 µg/ml.

**Determination of MIC and growth rates.** Minimum inhibitory concentrations were determined using a modified version of the broth microdilution method[67]. Briefly, an overnight culture of *E. coli* was diluted to ~10$^7$ cfu/ml in a final volume of 200 µl M9 medium with increasing concentrations of antibiotic (two-fold steps) or BAC in 96-well microplates (polypropylene (PP), Greiner Bio One) and cultivated at 37 °C for 24 h with continuous linear agitation in a BioTek Epoch 2 microplate reader. Optical density measurements in a BioTek Epoch 2 microplate reader were collected with the manufacturer software Gen5 3.09. The BAC concentrations in the MIC and MBC assays were 5, 10, 15, 20, 30, 40, 60, 100, 200 µM. Growth rates were calculated from OD$_{600}$ measurements taken every 5–10 min from cultures growing in M9 with and without antimicrobials, using a previously described algorithm implemented in Python 3.6[68]. We used 96-well microplates made from PP instead of the commonly used 96-well microplates from polystyrene (PS) to avoid adsorption of cationic agents (BAC, colistin) to the negatively charged surface of PS.

**Determination of tolerance through time-kill assays.** We developed a protocol that allows us to reproducibly follow the fast kill kinetics of BAC in up to six parallel cultures with high temporal resolution (down to 30 s between timepoints). Time kill assays were conducted in 2 ml tubes (PP, LABSOLUTE) in a total volume of 900 µl at 37 °C with agitation (1200 rpm) in a tabletop Thermomixer (STAR-LAB). Sampling was done with a 12-channel pipette (STARLAB) with only every second tip holder equipped with a pipette tip to accommodate sampling directly from the tubes. Samples of 10 µl were taken and serially diluted in 96-well microplates with 90 µl PBS per well. From each well, 10 µl were spotted on square LB agar plates. For certain timepoints, the remaining 90 µl from the highest ($t_{0min}$) or lowest ($t_{10min}$, $t_{20min}$) dilution were plated on a round agar plate to increase the accuracy and decrease the detection limit for surviving cells. Plates were left to dry and incubated at 30 °C for 16–24 h, followed by an enumeration of colony-forming units (cfu).

Bacteria were inoculated into 10 ml M9 to a defined number of cells (10$^4$ cfu/ml) and incubated to an OD of 0.1–0.2 (exponential phase) or for 24 h (stationary phase) at 37 °C with agitation at 220 rpm. After 24 h, OD$_{600}$ was determined and adjusted to an OD$_{600}$ of 0.01 (~10$^7$ cfu/ml) or 1 (~10$^9$ cfu/ml) in spent medium. Except for the data in Fig. 1a, all time-kill assays were conducted with *E. coli* in stationary phase. To maintain the cellular physiology in the stationary phase, time-kill assays were conducted in spent medium from the pre-culture, which was obtained by removing cells from the pre-culture medium by centrifugation for 2 min at 16000×*g*. After dilution of the cells, 10 µl were sampled to determine the initial cell concentration. BAC was added to a final concentration of 60 µM (~21.2 µg/ml) and samples were taken at short time intervals for a total of 20 min. To determine the fraction of survivors in the antibiotic persister strains, the initial time point before the addition of BAC and the final time point after 20 min of BAC exposure were sampled. Survival fractions were scaled to the survival fraction of the corresponding wild-type strain. For the time-kill assays in Fig. 1d, samples were taken after 5, 10, and 20 min.

To determine the tolerance against antibiotics, stationary cells were diluted to 10$^9$ cfu/ml in fresh M9 medium containing glucose and antibiotic at several-fold the MIC and survival were determined by serial dilution and plating after 5 h of exposure. The concentrations of the antibiotics were: ampicillin 100 µg/ml (MIC: 2–4 µg/ml), ciprofloxacin 1 µg/ml (MIC: 0.005–0.01 µg/ml), colistin 10 µg/ml (MIC: 0.25–0.5 µg/ml), gentamicin 7.5 µg/ml (MIC: 0.25–0.5 µg/ml).

**Determination of time-kill parameters.** To determine time-kill parameters and to distinguish between unimodal and bimodal killing kinetics, a model was fitted to the time-kill data in Fig. 1a, b. The equation models the time evolution of a population of cells C, consisting of two sub-populations of cells with initial cell numbers $N_0$ and $P_0$, which are killed with rates $k_n$ and $k_p$:

$$C(t) = N_0\,e^{-k_n t} + P_0\,e^{-k_p t}$$

If only one subpopulation exists, resulting in unimodal kill kinetics, $P_0$ becomes zero and the second part of the sum is eliminated. Both models were fit to the data and the superior model (unimodal vs bimodal) was selected based on Akaike information criterion (AICc) corrected for small sample sizes[69]. The data was weighted by the standard error of the colony counts.

**Evolution experiment.** In the evolution experiment, bacteria were grown in 10 ml M9 with glucose at 37 °C and agitation at 220 rpm. We increased the number of cells compared to the time-kill assays in Fig. 1 by a factor of 100 from 10$^7$ cfu/ml to 10$^9$ cfu/ml to increase the chance for mutations to occur and to minimize genetic drift. Killing by BAC is subject to the inoculum effect and as such cell-density dependent[40,70]. Thus, we increased the BAC concentration to achieve a reduction in the viable cell count by a factor of 10$^4$–10$^5$ after 15 min of exposure, similar to the time-kill assays in Fig. 1. After 24 h, samples were taken, adjusted to ~10$^9$ cfu/ml in spent medium, and treated with 120 µM BAC for 15 min in a total volume of 900 µl. After treatment, 100 µl from each tube were diluted 1:100 in 10 ml of fresh M9 to restart the cycle. Survivors were monitored by plating 100 µl from each flask on LB agar plates, with serial dilution in PBS when appropriate. Daily glycerol stocks were taken to be able to restart the experiment in case of contamination.

To control for adaptation to the experimental conditions, we took six replicate cultures along, which were subjected to the same protocol except that water was added instead of BAC. Control lines were diluted at the same factor (1:100) as the treated lines for two reasons: first, to avoid genetic drift, which can result in the selection of random and thus difficult to interpret mutations in the control lines, masking mutations that are related to adaptation to the evolution protocol. Second, to ensure the passage of the tolerant subpopulation we observed in the killing assays, which would be diluted out by higher dilution factors. While this resulted in less cumulative generations for the control lines due to the absence of killing it should be noted that the difference in the cumulative number of cell divisions after 15 rounds between treated and control lines was estimated to be only approx. 10% (3.2 × 10$^{10}$ divisions versus 2.9 × 10$^{10}$ divisions; treated versus control, respectively). Over the course of the experiment, the control lines showed a less than four-fold increase in the survival fraction compared to the first round.

After the evolution experiment, glycerol stocks were streaked on LB agar to obtain single colonies, and two random colonies per line were selected for phenotypic and genotypic analyses. Colonies were selected from all six treated lines and from two control lines.

**Statistical testing.** Statistical testing was done in Python 3.8, using the `scipy.stats` module. Data obtained in the growth rate assays, biofilm assays, surface charge assays, and motility assays were assumed to be normally distributed. Data obtained in the survival assays were assumed to be log-normally distributed and subjected to log-transformation prior to statistical testing.

**Sequencing and variant calling.** Genomic DNA from individual clones and evolved populations were isolated directly from 350 µl glycerol stocks from stationary phase cultures, using the peqGOLD bacterial DNA mini kit (VWR Peqlab).

Genome sequencing of individual clones was provided by MicrobesNG (http://www.microbesng.uk) which is supported by the BBSRC (grant number BB/L024209/1). Illumina sequencing yielded 2 × 250bp paired-end reads with ≥30-fold average coverage. No mutations were detected in the control lines. In addition to the individual clones, we sequenced all six treated populations and two control populations to confirm that no rare mutants were selected for sequencing and phenotyping. Mixed populations were sequenced by EuroFins (Germany) with ~300-fold coverage (2 × 150bp paired-end reads). Population sequencing revealed that all isolated mutants were at or close to fixation with mutation frequencies in the population between 0.72 and 1.

Sequence alignment and variant calling were done with breseq-0.33[71] against *E. coli* K12 MG1655 reference sequence (NCBI RefSeq accession: NC_000913.3; RefSeq assembly accession: GCF_000005845.2). Breseq was run in consensus mode on sequencing data from individual clones and in polymorphism mode on sequencing data from mixed populations to extract mutations with frequencies between 0.05 and 1.

**Modeling population dynamics.** We modeled the population dynamics during the evolution experiment using a system of ordinary differential equations (ODE). In the model, two genotypes (wild-type and mutant) compete for a common substrate. For each genotype, maximal growth rate, surviving fraction upon BAC challenge, and yield (cells per unit substrate) can be set. The ODE system that describes the change of the number of bacterial cells, *N*, and the resource, *R*,

consists of the following equations:

$$\frac{\mathrm{d}N}{\mathrm{d}t} = \sum_i^n G_i \mu_i$$

$$\frac{\mathrm{d}R}{\mathrm{d}t} = -\sum_i^n G_i \mu_i e_i$$

With $\mu_i = \mu\max_i \frac{R}{R+K_m}$, $G_i$ as the $i$th genotype, $e_i$ the genotype-specific inverse yield (glucose/cells) and the Monod-constant $K_m$ which was set to 0.25. We developed a Python package (https://github.com/nnordhol/ODEvolution [72]) to simulate the population dynamics during experimental evolution in batch culture. To simulate the population dynamics in the evolution experiment in Fig. 2, the ancestor started with $10^4$ cfu/ml and the mutant with $10^{-1}$ cfu/ml (i.e., $10^5$ ancestor cells and 1 mutant in a total culture volume of 10 ml). Yield was inferred from cfu to $OD_{600}$ conversion factors of the individual strains. Each growth cycle went on for 24 h after which a killing factor and a dilution factor were applied. Population growth seizes upon exhaustion of the substrate, and a genotype-specific factor is applied to simulate killing by BAC. After applying a common dilution factor (1:100), another round of growth was started by setting the value for the common resource to the initial value (20 mM). A genotype is considered extinct when there is less than one cell left after killing and dilution, and the number of cells of this genotype is set to 0.

The software package (https://github.com/nnordhol/ODEvolution) has two types of objects: experiments and genotypes. Parameters of experiments are set by experimental design (culture volume, passaged volume, resource concentration, time between rounds of killing). The parameters of genotypes can be determined experimentally (growth rate, surviving fraction, yield).

**Determination of surface charge changes with cytochrome c**. Changes in the surface charge using cytochrome c were determined by a method described previously[73,74]. Evolved clones and the ancestor strain were grown for 24 h in M9, harvested by centrifugation (8 min, 4000×$g$), and washed twice in 10 mM KPi buffer (6.96 mM $K_2HPO_4$, 3.04 mM $KH_2PO_4$) with 0.01% BSA and adjusted to an $OD_{600}$ of 4.5. Aliquots of 1 ml were pelleted and resuspended in 500 µl KPi buffer with 0.01% BSA and 0.25 mg/ml cytochrome c (Sigma, C2506). After incubation for 15 min @ 37 °C with shaking, cells were removed by centrifugation and cytochrome c in the supernatant was quantified photometrically at a wavelength of 410 nm. $OD_{410}$ values were subtracted from $OD_{410}$ values of 0.25 mg/ml cytochrome c in KPi without cells to determine the amount of cytochrome c bound by the cells. Shown is the difference of cytochrome c bound by the cells relative to the ancestor.

**Biofilm assay**. Biofilm formation was assessed in 96-well PP microplates, based on a previously published protocol[75]. Cells were grown over night in M9 medium and diluted 1:100 in fresh M9 to a final volume of 100 µl. Cultures were incubated for 48 h at 37 °C without shaking. After removal of the supernatant, biofilms were washed twice with water and stained with 150 µl 0.1% crystal violet solution for 10 min at room temperature with shaking at 800 rpm. The crystal violet solution was removed, biofilms were washed thrice with water and then left to dry for 30 min. Crystal violet from stained biofilms was extracted by addition of 220 µl solvent solution (80% ethanol, 20% acetone) and incubation for 15 min at room temperature with agitation at 800 rpm. One hundred fifty µl of the solution was transferred to a new 96-well microplate and the optical density at 570 nm was determined.

**Motility assay**. Soft agar M9 plates containing 0.3% agar were inoculated by dropping 2.5 µl ($10^7$ cfu) of a stationary overnight culture onto the plate. After evaporation of the droplet, plates were covered with a wetted black felt cloth and the plates were closed. The felt cloth was wetted with water to prevent evaporation from the agar. Plates were incubated upright at 35 °C and imaged after 24 h with an Epson V370 flatbed scanner. The motility area was determined with FIJI 1.53c[76].

**Reporting summary**. Further information on research design is available in the Nature Research Reporting Summary linked to this article.

## Data availability
Source data for Figs. 1, 2, 4, and 5 are provided with this paper. The whole-genome sequencing data generated in this study have been deposited in the NCBI database under project ID PRJNA735069. The reference sequence of E. coli K12 MG1655 was obtained from the NCBI RefSeq database (RefSeq accession: NC_000913.3; RefSeq assembly accession: GCF_000005845.2). Source data are provided with this paper.

## Code availability
The Python code that implements the model is available at https://github.com/nnordhol/ODEvolution[72].

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

## Acknowledgements

This study was supported by the Federal Institute for Materials Research and Testing (BAM). Genome sequencing was provided by MicrobesNG (http://www.microbesng.uk) which is supported by the BBSRC (grant number BB/L024209/1). We thank R. Mutzel, K. Lewis, J. Michiels, K. Gerdes, C. Guet, T. Bergmiller and M.J. Dunlop for providing bacterial strains. We thank C. Guet for critically reading the manuscript. We thank members of the Schreiber and Koerdt labs at BAM for helpful discussions.

## Author contributions

F.S., N.N. and S.B.I.S. conceived and designed experiments. N.N., O.K. and S.B.I.S. conducted the experiments. N.N., F.S. and O.K. analyzed the data. N.N. and F.S. wrote the manuscript. All authors read and approved the final version of the manuscript.

## Funding

## Competing interests

The authors declare no competing interests.
