## [Peer Review File · Nature Communications]

Persistence against benzalkonium chloride promotes rapid evolution of tolerance during periodic disinfectionEditorial Note: Parts of this Peer Review File have been redacted as indicated as we could not obtain permission to publish the reports of reviewer #2.

REVIEWER COMMENTS

Reviewer #1 (Remarks to the Author):

The authors explore the existence of populations of persisters to BAC in E.coli, and found that they exist, they can be enriched upon periodic selection and they can render antibiotic tolerance. Later on, they describe mutations involved in persistence, some of them already described in the case of antibiotics, but also new ones in lpxM. The main authors' statement is that persistence to biocides occurs in a similar way than resistance to antibiotics and that this persistence increases tolerance to antibiotics.

The fact the persistence is not antibiotic-specific is well established in the field. Persistence is a situation of non-susceptibility to different injuries; in this respect the finding that persistence to biocides present the same characteristics as persistence to antibiotics, and that there exists a co-persistence phenotype is what could be expected; not particularly novel. Both biocides and antibiotics are antimicrobials. The main specificity concerning biocides is that their toxicity precludes its use to treat infections and that they usually interact with cell membranes something that is not so common for antibiotics, although polymyxins also target cell membrane.

To sum up, the study is a transfer of concepts well established in the field of antibiotic resistance to the biocides field. In this regard, the article is not conceptually novel, although the results merit to be communicated in a specialized journal.

Specific comments:

Lines 52-53: The authors would like discussing that the acquisition by HGT of QAC resistance genes is also relevant.

Lines 57-58: The authors stater that "Disinfectants are typically applied transiently and at concentrations above the MIC. Therefore,selection pressure in disinfection is on transient survival, i.e. tolerance" This statement is misleading. Any antimicrobial (biocides or antibiotics) is transiently applied: bacteria are not under constant selective pressure and tolerance is not the consequence of transient selection: tolerance is a transient phenotype, which can be enriched when bacteria are confronted with periodic antimicrobial selective pressure.

Lines 392-396: There are also reports where mutants are straightforward selected in plates containing inhibitory concentrations of biocides.

Reviewer #2 (Remarks to the Author):

[Redacted]

Reviewer #3 (Remarks to the Author):

The manuscript „Persistence against benzalkonium chloride promotes rapid evolution of tolerance under periodic disinfection” provides new and important information about the formation of persisters during disinfection. The experiments have performed carefully and data correctly interpreted.

Still, the interpretation of antibiotic persister genes is too far reaching (lines 172-173, 379-383). Three of the genes tested are overexpressed toxins. There is currently discussion what the toxin overexpression means (PMID: 33177189). In fact, it has been shown that overexpression of any toxic protein increases the level of persisters (PMID: 16672603). Therefore overexpressed toxins cannot be considered as specific antibiotic persister genes.

Lines 425-431. Why was it needed to adapt the strain to M9 medium? Moreover, lines 429-431 state that the adaptation effort did not result in any mutations. Therefore there was no adaptation.

Line 473 “PP over plates from polystyrene (PS)”. This is confusing statement. PS plates covered with PP?

Lines 370-371 please check grammar (“kinetics are caused by a persister subpopulation”). Certainly not all the killing kinetics is determined by persisters.

We would like to thank the reviewers for taking the time to provide valuable comments on our manuscript. We carefully addressed all comments. By doing this we now can present a substantially improved manuscript. Textual changes to the manuscript are indicated below the reply to each specific comment and are highlighted in the revised version of the manuscript.

REVIEWER COMMENTS

Reviewer #1 (Remarks to the Author):

The authors explore the existence of populations of persisters to BAC in *E.coli*, and found that they exist, they can be enriched upon periodic selection and they can render antibiotic tolerance. Later on, they describe mutations involved in persistence, some of them already described in the case of antibiotics, but also new ones in lpxM. The main authors' statement is that persistence to biocides occurs in a similar way than resistance to antibiotics and that this persistence increases tolerance to antibiotics.

The fact the persistence is not antibiotic-specific is well established in the field. Persistence is a situation of non-susceptibility to different injuries; in this respect the finding that persistence to biocides present the same characteristics as persistence to antibiotics, and that there exists a co-persistence phenotype is what could be expected; not particularly novel. Both biocides and antibiotics are antimicrobials. The main specificity concerning biocides is that their toxicity precludes its use to treat infections and that they usually interact with cell membranes something that is not so common for antibiotics, although polymyxins also target cell membrane.

To sum up, the study is a transfer of concepts well established in the field of antibiotic resistance to the biocides field. In this regard, the article is not conceptually novel, although the results merit to be communicated in a specialized journal.

Reply:

We thank the reviewer for the insightful comments which helped to improve the quality of the manuscript. We now include more details and clearer definitions in the introduction and a more balanced discussion.

Specific comments:

Reviewer's comment:

Lines 52-53: The authors would like discussing that the acquisition by HGT of QAC resistance genes is also relevant.

Reply:

We thank the reviewer for this comment. It made us aware that our introduction missed some information on known mechanisms for reduced QAC susceptibility (including those promoted by

HGT). We added information on HGT, as suggested by the reviewer, and, in addition, information on mechanisms acquired through mutations.

Change to the manuscript:

L49-56: Several studies showed examples of reduced susceptibilities to QACs, occurring in natural, clinical and industrial isolates and, to a lesser extent, in laboratory evolution experiments^{9,14-16}. Reduced susceptibility to QACs is underpinned by acquiring mutations in genes that increase QAC efflux by upregulation of inherent multidrug-efflux pumps¹⁷ or by acquiring specialized QAC efflux pumps via horizontal gene transfer^{18,19}. In addition, strains that have been evolved towards decreased susceptibility show reduced expression of porins related to reduced QAC uptake^{20,21} and changes in membrane structure or composition^{21,22}. QAC resistance mechanisms can confer cross-resistance to antibiotics.

Reviewer's comment:

Lines 57-58: The authors state that "Disinfectants are typically applied transiently and at concentrations above the MIC. Therefore, selection pressure in disinfection is on transient survival, i.e. tolerance" This statement is misleading. Any antimicrobial (biocides or antibiotics) is transiently applied: bacteria are not under constant selective pressure and tolerance is not the consequence of transient selection: tolerance is a transient phenotype, which can be enriched when bacteria are confronted with periodic antimicrobial selective pressure.

Reply:

We thank the reviewer for pointing out the ambiguity and that our paragraph left the impression that antibiotics are not applied transiently and exert a constant selective pressure. Indeed, both, antibiotics and disinfectants, are transiently applied and there is no constant selective pressure. The current literature on antibiotics suggests that evolution for increased survival (defined as 'tolerance') is related to periodic, lethal exposure and precedes resistance evolution. In this paragraph, our aim was to make the point that this is also the case for disinfectants, not that disinfectants are distinct from antibiotics. We re-phrased the section to avoid this impression. Secondly, we more clearly defined the terms 'resistance' and 'tolerance', including the reviewer's notion of defining tolerance as a phenotype. Tolerance phenotypes are, however, underpinned by genetic mechanisms, making tolerance accessible to evolution.

Change to the manuscript:

L64-71: Disinfectants and antibiotics are typically applied periodically and, in many cases, at lethal concentrations. Periodic exposure to lethal concentrations of antibiotics has been shown previously to exert a strong selective pressure on increased survival, leading to the selection for tolerance^{25,26}. Tolerance has been defined as the ability to *survive* transient exposure to an antimicrobial at otherwise lethal concentrations²⁷, including constitutive and inducible tolerance phenotypes. The tolerance phenotype is underpinned by genetic mechanisms, making tolerance accessible to evolution. Tolerance against antibiotics can act as a stepping stone for evolution of resistance²⁸, which is defined as the ability to *grow* at high concentrations of an antimicrobial.

Reviewer's comment:

Lines 392-396: There are also reports where mutants are straightforward selected in plates containing inhibitory concentrations of biocides.

Reply:

We have changed the paragraph to reflect the complementary nature of the different approaches when assessing the risk of adaptation to biocides. One approach is the stepwise increase of biocide concentration (including serial dilution at sub-inhibitory concentrations or selection on plates with inhibitory concentrations). The other approach is the one applied by us.

Change to the manuscript:

L409-417: The rapid evolution of BAC tolerance in our experiment highlights the need for complementary approaches when assessing the potential for evolution of bacterial defense mechanisms to biocides. These approaches should include (i) traditional experiments selecting for growth during adaptation to increasing concentrations of biocides (including serial dilution at sub-inhibitory concentrations or selection on plates with inhibitory concentrations) and subsequent assessment of the MIC, and (ii) experiments selecting on increased transient survival, as performed in this study. Both approaches will be important to fully capture the risk associated with periodic failure of disinfection. It will be interesting to study whether the evolution of tolerance facilitates the evolution of high-level resistance to biocides, as was shown previously for antibiotics²⁸.

Reviewer #2

[Redacted]

Reviewer #3 (Remarks to the Author):

The manuscript „Persistence against benzalkonium chloride promotes rapid evolution of tolerance under periodic disinfection” provides new and important information about the formation of persisters during disinfection. The experiments have performed carefully and data correctly interpreted.

Reply:

We thank the reviewer for the insightful comments that resulted in an improved version of the manuscript. As requested, we have clarified terminological and methodological ambiguities.

Reviewer's comment:

Still, the interpretation of antibiotic persister genes is too far reaching (lines 172-173, 379-383). Three of the genes tested are overexpressed toxins. There is currently discussion what the toxin overexpression means (PMID: 33177189). In fact, it has been shown that overexpression of any toxic protein increases the level of persisters (PMID: 16672603). Therefore overexpressed toxins cannot be considered as specific antibiotic persister genes.

Reply:

We understand that the term “antibiotic persister genes” might be misleading, as it implies that the main function of these genes is persister formation. The term was used for brevity to describe genes whose deletion or overexpression has been associated with changes in the fraction of antibiotic persisters. We did several changes throughout the text to clarify that the genes are not specific antibiotic persister genes.

Change to the manuscript:

L166-168: To this end, we screened mutants of genes that have been associated with changes in the fraction of antibiotic persisters for their ability to survive BAC treatment.

L157-158: Overlap between mechanisms that generate persisters against BAC and antibiotics

L190-192: Taken together, our data show that there is an overlap in the mechanisms that generate persisters against antibiotics and BAC. This suggests that the BAC persister subpopulation is partly composed of antibiotic persisters.

L395-397: Second, describing the phenomenon of BAC persistence enabled us to link the mechanism of BAC survival to mechanisms that are known to generate antibiotic persisters.

Reviewer's comment:

Lines 425-431. Why was it needed to adapt the strain to M9 medium? Moreover, lines 429-431 state that the adaptation effort did not result in any mutations. Therefore there was no adaptation.

Reply:

The adaptation was done to allow the occurrence of mutations that are specific to the culture conditions, i.e. M9 medium, in order to prevent false interpretation of such mutations as an adaptation to the BAC treatment. Considering that metabolic genes have previously been shown to play a role in persister formation, we deemed this adaptation to be necessary. While there were no adaptive mutations, we think it is important to mention this procedure, as it shows that adaptive mutations to the medium do not easily occur, so they don't mask the adaptive response to BAC.

Change to the manuscript:

L443-445: Culturing the ancestor in M9 glucose by serial passage for 150 generations was done to allow adaptive mutations to the culture medium but did not result in additional mutations. In the main text, we refer to the strain pre-cultured in M9 as the wild type or the ancestor.

Reviewer's comment:

Line 473 "PP over plates from polystyrene (PS)". This is confusing statement. PS plates covered with PP?

Reply:

We agree that this statement was ambiguous. We have changed the sentence to clarify its meaning and taken the chance to unify the terminology (microplate throughout the text).

Change to the manuscript:

L486-488: We used 96-well microplates made from PP instead of the commonly used 96-well microplates from polystyrene (PS) to avoid adsorption of cationic agents (BAC, colistin) to the negatively charged surface of PS.

Reviewer's comment:

Lines 370-371 please check grammar ("kinetics are caused by a persister subpopulation"). Certainly not all the killing kinetics is determined by persisters.

Reply:

We have rephrased the sentence to a more direct statement of our findings.

Change to the manuscript:

L387: Our work provides evidence that *E. coli* forms persisters against BAC, a widely used disinfectant.

REVIEWERS' COMMENTS

Reviewer #1 (Remarks to the Author):

Although I still think that the article does not have the level of novelty required for its publication in Nature Communications and it would be best suited for a more specialized journal, this is an editorial decision that I can agree with, without any problem.

Concerning more technical issues, the authors rightly addressed all my queries.

Reviewer #2 (Remarks to the Author):

[Redacted]

Reviewer #3 (Remarks to the Author):

The manuscript has been appropriately revised. I recommend to accept it for publication.